# Neural Tangent Knowledge Distillation for Optical Convolutional Networks

**Jinlin Xiang**[*‡]    **Minho Choi**[*‡¶]    **Yubo Zhang**[‡]    **Zhihao Zhou**[‡]    **Arka Majumdar**[‡§∥]

**Eli Shlizerman**[†‡∥]

## Abstract

Hybrid Optical Neural Networks (ONNs, typically consisting of an optical frontend and a digital backend) offer an energy-efficient alternative to fully digital deep networks for real-time, power-constrained systems. However, their adoption is limited by **two main challenges**: the accuracy gap compared to large-scale networks during training, and discrepancies between simulated and fabricated systems that further degrade accuracy. While previous work has proposed end-to-end optimizations for specific datasets (e.g., MNIST) and optical systems, these approaches typically lack generalization across tasks and hardware designs. To address these limitations, we propose a task-agnostic and hardware-agnostic pipeline that supports image classification and segmentation across diverse optical systems. To assist optical system design before training, we design the metasurface layout based on fabrication constraints. For training, we introduce Neural Tangent Knowledge Distillation (NTKD), which aligns optical models with electronic teacher networks, thereby narrowing the accuracy gap. After fabrication, NTKD also guides fine-tuning of the digital backend to compensate for implementation errors. Experiments on multiple datasets (e.g., MNIST, CIFAR, Carvana Image Masking Dataset) and hardware configurations show that our pipeline consistently improves ONN performance and enables practical deployment in both pre-fabrication simulations and physical implementations.

## 1 Introduction

Optical Neural Networks (ONNs) offer a promising approach to achieve efficient computation and energy use compared to digital implementations such as Convolutional Neural Networks (CNNs) and Vision Transformers (ViTs), making them well-suited for resource-constrained, real-time physical systems [1]. For example, ONNs have been proposed for power-limited applications (illustrated in Figure 1.a), including satellites [2], unmanned aerial vehicles [3], smart home devices [4], autonomous driving systems [5], wearable electronics [6], and medical devices [7].

Among different ONN implementations, hybrid optical-electronic architectures are practical options under current hardware constraints [8]. In such systems, the optical frontend accelerates computation at the speed of light, while the digital backend refines predictions to improve robustness [9]. The

---

[*]These authors contributed equally to this work.

[†]Department of Applied Mathematics, University of Washington, Seattle, WA 98195, USA.

[‡]Department of Electrical and Computer Engineering, University of Washington, Seattle, WA 98195, USA.

[§]Department of Physics, University of Washington, Seattle, WA 98195, USA.

[¶]Department of Electrical Engineering, Ulsan National Institute of Science and Technology, Ulsan 44919, South Korea

[∥]Corresponding authors: arka@uw.edu, shlizee@uw.edu

39th Conference on Neural Information Processing Systems (NeurIPS 2025).

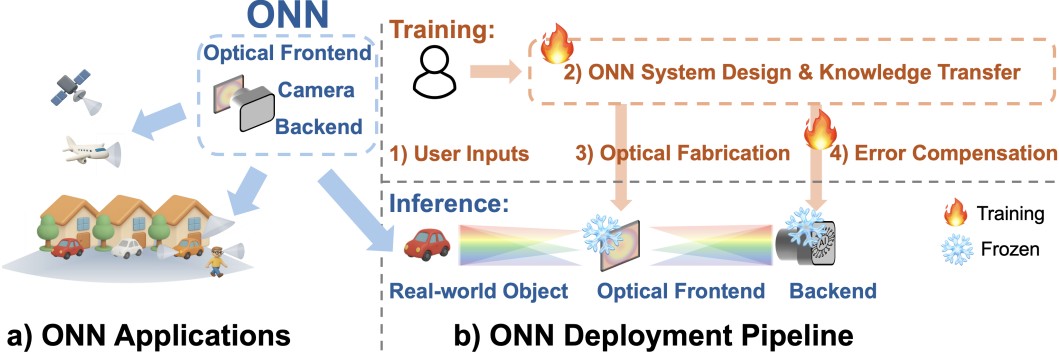

**a) ONN Applications**    **b) ONN Deployment Pipeline**

Figure 1: Overview of potential applications for ONNs and our proposed deployment pipeline. (a) ONNs for real-time decision-making in power-constrained scenarios. (b) Our proposed pipeline includes user-driven design, knowledge transfer training, fabrication, and error compensation.

optical frontend generally consists of a single linear layer (as shown in Figure 1.a), since: (1) implementing nonlinear activation functions in physical optics remains extremely challenging due to material and device limitations; and (2) without nonlinearity, multiple linear transformations can be mathematically compressed into a single linear transformation. Moreover, from a theoretical standpoint, the universal approximation property could still be satisfied by shallow networks, suggesting that hybrid ONNs retain sufficient expressive power for a wide range of tasks when appropriately optimized [10–12].

Despite their promises, ONNs remain difficult to design and train due to both **architectural limitations** and **fabrication-related challenges**. First, existing ONN architectures are typically significantly simpler than modern deep CNNs or ViTs. These simplifications cannot be directly obtained through pruning or quantization [13–15]. Second, physical fabrication and experimental deployment inevitably introduce various sources of noise, such as optical misalignment, material variability, and measurement noise, further degrading performance [16]. While some end-to-end optimization strategies have been proposed to address these challenges, they are typically designed for a specific dataset (e.g., MNIST) and tailored to a particular optical system, rather than providing a generalized solution (as also summarized in Related Works). In contrast, we aim to develop a task-agnostic and hardware-agnostic pipeline that can generalize across different datasets and optical hardware setups.

To address these challenges, knowledge transfer, particularly Knowledge Distillation (KD), offers a promising solution by transferring knowledge from pre-trained digital networks to optical models [17, 18]. Moreover, recent work shows that successful KD implicitly leads to student-teacher Neural Tangent Kernel (NTK) similarity, where NTK captures how the network's predictions change with respect to small changes in its parameters [19]. As we show here, utilizing NTK for matching is particularly effective for ONNs, as the NTK provides a linear approximation of network behavior, naturally aligning with the linear operations performed by optical systems.

Thus, we propose a Neural Tangent Knowledge Distillation (NTKD) pipeline that generalizes across different optical network designs and datasets to support multiple tasks such as classification and segmentation (also shown in Figure 1.b). The pipeline starts with specifying the task, the dataset, and the optical structure. Then, NTKD optimization transfers knowledge from digital teacher models to hybrid ONNs by matching their NTKs, effectively transferring the relational structure between classes rather than just matching final predictions. Furthermore, the pipeline compensates for errors introduced during fabrication and experimental deployment by aligning the student's and teacher's NTKs through a small fraction (e.g., 10%) of real experimental data.

In summary, our contributions are as follows:

• We introduce a Neural Tangent Knowledge Distillation (NTKD) pipeline that supports diverse tasks and optical structures, addressing the challenges of shallow architectures and physical imperfections.

• We experimentally validate our pipeline with different ONN implementations on both classification and segmentation tasks, demonstrating its effectiveness through both simulations and fabrications.

Table 1: Summary of previous ONN works categorized by task type (classification, segmentation) and implementation level—either simulation only (denoted as **Sim**) or with physical fabrication and experimental validation (denoted as **Fab**).

| ONN Capability | Works | Classification (Sim) | Classification (Fab) | Segmentation (Sim) | Segmentation (Fab) |
|---|---|:---:|:---:|:---:|:---:|
| **Monochromatic** | 2018–2025: [9, 16, 20–27, 30–38] | ✓ | ✓ | ✗ | ✗ |
| **Polychromatic** | 2023–2025: [16, 28, 29, 39, 40] | ✓ | ✓ | ✗ | ✗ |
| | ExtremeMETA (2025) [5] | ✗ | ✗ | ✓ | ✗ |
| | **Ours (NTKD)** | ✓ | ✓ | ✓ | ✓ |

• We leverage NTK analysis to estimate the achievable accuracy of given hybrid ONNs, providing theoretical guidance on their design and optimization.

## 2 Related Works

**ONN Tasks and Implementations:** Table 1 categorizes ONN applications into two tasks (classification and segmentation) and two optical implementations (monochromatic and polychromatic systems). Most previous work focused on monochromatic ONNs for MNIST image classification, including fully optical systems that performed linear transformations [20, 21], physically nonlinear ONNs that used atomic vapors or intensifiers [22–24], and hybrid architectures that combined an optical frontend with a digital backend [25–27]. Previous polychromatic ONNs for classification were limited to small datasets such as CIFAR-10, as ONN architectures faced challenges in scaling to complex benchmarks [28, 29]. Segmentation tasks are still in the early stages, with a previous study based only on simulation [5]. Our work considers both classification and segmentation tasks. In our pipeline, image reconstruction is implicitly incorporated by encouraging the optical frontend output to align with the simulated result, as the physical output deviates from simulation and requires correction.

**Transfer learning for ONNs:** Transfer learning, particularly Knowledge Distillation (KD), offers a promising solution for transferring knowledge from pre-trained digital networks to optical models [17, 41]. KD minimizes the divergence between a compact student model's predictions and those of a pre-trained teacher model, thereby encouraging the student to actively mimic the teacher's behavior [17, 42]. Beyond conventional KD, recent studies have shown NTK-based approaches to understand knowledge transfer [43]. For example, theoretical insights into KD transfer risk and data efficiency in wide networks have been established through NTK analysis [44]. Subsequent work demonstrated that successful knowledge distillation implicitly led to student-teacher NTK alignment [19], and NTK similarity was further applied to quantify task affinities in multi-task learning [45–47]. In contrast, our work targets physically constrained ONNs, and introduces an explicit NTK matching strategy to directly guide the distillation process from a digital teacher to an optical student.

**Compensation Strategies for Practical ONNs:** Fabrication imperfections and system noise in physical ONNs often lead to significant performance drops compared to simulations. Some approaches used deep learning to model the system directly in a data-driven manner [48], including ONN auto-learning [49, 50], where ONNs were trained to fit experimental input-output mappings. Other methods introduced physical information via hardware-in-the-loop training. For example, physics-constrained frameworks embedded fabrication-aware models and losses to better align learning with optical behavior [16]. Moreover, some approaches avoided simulation entirely by randomly fabricating optical kernels and training a digital backend to adapt to the fixed frontend structure [51–55]. This simplified fabrication but placed the learning burden entirely on the backend. It is also not clear if such random surfaces perform better than an ordinary lens. In contrast, our work identifies sources of physical errors and introduces an NTK alignment strategy for effective compensation.

## 3 Methods

### 3.1 Optical Frontend Design

At the initialization of the pipeline, user inputs are required to define the optical system (also shown in Figure 2.1). Specifically, the user specifies (1) the physical size of the optical frontend (e.g., the

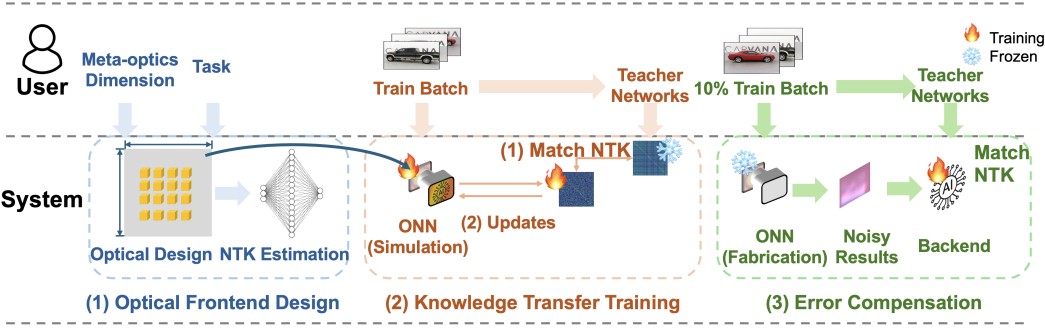

Figure 2: Overview of the pipeline. It consists of three steps: (1) Optical Frontend Design based on user-specified inputs, (2) Knowledge Transfer Training using Neural Tangent Kernel (NTK) matching, and (3) Error Compensation for fabricated optical frontends.

number of meta-optic kernels), (2) the target dataset for the task, and (3) the desired network structure, such as the number of layers and channels. Prior works have demonstrated that optical convolution can be physically realized using either a 4f system or a PSF-based free-space propagation system [18, 30, 38, 56]. In this work, we practically implement a PSF-based metasurface design due to its advantages in compactness, alignment robustness, and ease of fabrication [29, 30].

**Optical Frontend Layout:** We consider a metasurface of size $(h, w)$, onto which we aim to place $n_{kernels}$ square optical kernels, each of size $k$ (in mm), with a minimum edge-to-edge spacing $d$ to satisfy fabrication constraints. To compute the maximum number of kernels that can be placed while preserving symmetry, we define

$$n_{\text{cols}} = \left\lfloor \frac{w - d}{k + d} \right\rfloor, \quad n_{\text{rows}} = \left\lfloor \frac{h - d}{k + d} \right\rfloor, \quad n_{kernels} = n_{\text{cols}} \times n_{\text{rows}}. \tag{1}$$

**Performance Estimation:** Once the physical layout of the ONN is determined, we aim to estimate its expected performance without empirical training. We adopt the Neural Tangent Kernel (NTK) framework, which captures the training dynamics of **infinitely wide neural networks** under gradient descent. In particular, we introduce a reference network that shares the same architecture as the designed ONN (e.g., number of layers and connectivity) but has infinite width at each layer. Under this assumption, the predictions of the reference network correspond to NTK regression [57, 58]. Let the reference network $f(x; \theta)$ be a neural network parameterized by $\theta$, which maps an input $x$ to an output $f(x; \theta)$. The NTK is defined as

$$\Theta(x, x') = \nabla_\theta f(x; \theta)^\top \nabla_\theta f(x'; \theta), \tag{2}$$

where $\nabla_\theta f(x; \theta)$ is the Jacobian of the network output with respect to its parameters. Let $\{x_i^{\text{train}}, y_i^{\text{train}}\}_{i=1}^{n_{\text{train}}}$ be the training data and $\{x_i^{\text{test}}\}_{i=1}^{n_{\text{test}}}$ be the test data. We compute

$$\Theta_{\text{train,train}} = \Theta(x^{\text{train}}, x^{\text{train}}) \in \mathbb{R}^{n_{\text{train}} \times n_{\text{train}}}, \quad \Theta_{\text{test,train}} = \Theta(x^{\text{test}}, x^{\text{train}}) \in \mathbb{R}^{n_{\text{test}} \times n_{\text{train}}}. \tag{3}$$

and use kernel regression to predict outputs on the test set

$$f(x^{\text{test}}; \theta) = \Theta_{\text{test,train}} \left( \Theta_{\text{train,train}} + \lambda I \right)^{-1} y^{\text{train}}, \tag{4}$$

where $\lambda$ is a regularization parameter, which is selected via grid search on a validation set.

The NTK-based performance estimation serves as a diagnostic tool to evaluate whether the specified ONN architecture is expressive enough for the given task. While this estimation is not used for training or loss computation, it provides an early signal to guide architectural decisions and allows users to iteratively refine the optical design before full training and fabrication. For example, if the estimated test accuracy is much lower than the expected performance, it may suggest a mismatch between the ONN's capacity (e.g., depth) and task complexity.

## 3.2 Knowledge Transfer Training

After the user specifies the ONN architecture, we train the system for the target task. We define a supervised learning problem with input-output pairs $(x, y)$, where $x$ represents the input samples and $y$ denotes the corresponding ground-truth labels. The network parameters $(\theta)$, including the optical frontend and the digital backend, are initialized and optimized jointly (shown in Figure 2.2).

**End-to-end loss:** The first loss term we consider is a standard end-to-end supervision loss, which directly minimizes the discrepancy between the network's predictions and the ground-truth labels. Formally, we optimize the following objective

$$\mathcal{L}_{\text{E2E}} = \mathbb{E}_{(x,y)\sim\mathcal{D}} \left[ \ell(f_{ONN}(x;\theta), y) \right], \tag{5}$$

where $\ell(\cdot, \cdot)$ is a standard loss function such as cross-entropy for classification tasks, $f_{ONN}(x;\theta)$ denotes the output of the network with parameters $\theta$, and $\mathcal{D}$ represents the training dataset.

**Neural Tangent Knowledge Distillation (NTKD) Loss:** In addition to the standard end-to-end loss, we introduce a knowledge transfer loss based on Neural Tangent Kernel (NTK). Specifically, we assume access to a pretrained teacher network, such as LeNet for MNIST, AlexNet for CIFAR-10, or U-Net for image segmentation tasks.

Given a minibatch of input samples $\{x_i\}_{i=1}^{n_{\text{batch}}}$, we compute the Jacobian matrices of both the teacher network ($f_{\text{teacher}}$) and student ONN ($f_{ONN}$) with respect to their parameters ($\theta_{\text{teacher}}, \theta_{ONN}$)

$$J_{\text{teacher}} = \left[ \frac{\partial f_{\text{teacher}}(x_i)}{\partial \theta_{\text{teacher}}} \right]_{i=1}^{n_{\text{batch}}}, \quad J_{ONN} = \left[ \frac{\partial f_{ONN}(x_i)}{\partial \theta_{ONN}} \right]_{i=1}^{n_{\text{batch}}}. \tag{6}$$

The Jacobians $J_{\text{teacher}} \in \mathbb{R}^{n_{\text{batch}} \times p_{\text{teacher}} \times n_{\text{class}}}$ and $J_{ONN} \in \mathbb{R}^{n_{\text{batch}} \times p_{ONN} \times n_{\text{class}}}$ may differ in width depending on the number of trainable parameters in each network. Here, $p_{\text{teacher}}$ and $p_{ONN}$ denote the number of parameters in the teacher and ONN, respectively. Their corresponding NTK matrices,

$$\Theta_{\text{teacher}} = J_{\text{teacher}} J_{\text{teacher}}^\top, \quad \Theta_{ONN} = J_{ONN} J_{ONN}^\top, \tag{7}$$

are both of size $n_{\text{batch}} \times n_{\text{batch}}$. We define the NTKD loss by minimizing the discrepancy between the NTK matrices of the teacher network and the ONN (e.g., MSE)

$$\mathcal{L}_{\text{NTKD}} = \mathbb{E}_{\{x_i\}_{i=1}^{n_{\text{batch}}}\sim\mathcal{D}} \left[ \ell\left(\Theta_{\text{teacher}}, \Theta_{ONN}\right) \right]. \tag{8}$$

Then, we minimize a weighted sum of two losses, controlled by hyperparameters $\alpha$ and $\beta$

$$\min_\theta \left( \alpha \mathcal{L}_{\text{E2E}} + \beta \mathcal{L}_{\text{NTKD}} \right). \tag{9}$$

## 3.3 Error Compensation

The physical fabrication uses the optimized simulation parameters obtained through the process described in Section 3.2. The fabrication fixes the optical frontend, and only the digital backend remains tunable. Due to unavoidable fabrication and experimental errors, discrepancies arise between the designed and realized optical system. Given an input image $a$ and convolution kernel $k$, the ideal output is $y = a * k$. The fabricated optical convolution output with noise at location $(i, j)$ is

$$\tilde{y}_{i,j} = \alpha\beta \sum_{m=1}^{k_{size}} \sum_{n=1}^{k_{size}} a_{i+m-1,j+n-1} \left(k_{m,n} + \delta_{m,n}\right) + \epsilon_{i,j}. \tag{10}$$

Here, the scaling factors $\alpha$ (image brightness) and $\beta$ (image–kernel misalignment) can be experimentally calibrated to match the designed system, while the sensor noise $\epsilon$ is primarily determined by the imaging device characteristics. We further quantify the impact of fabrication noise $\delta$ (with proof provided in the Supplementary). In particular, we show that perturbations in the NTK caused by kernel fabrication errors ($\delta_{ij}$) scale as

$$\|\Delta\Theta_{ONN}\| \sim O\left(\frac{\|\delta\|}{m}\right), \tag{11}$$

where $\Delta\Theta_{ONN}$ denotes the NTK perturbation, and $m$ is the number of kernels (i.e., the network width). This result indicates that networks with more kernels are inherently more robust to fabrication

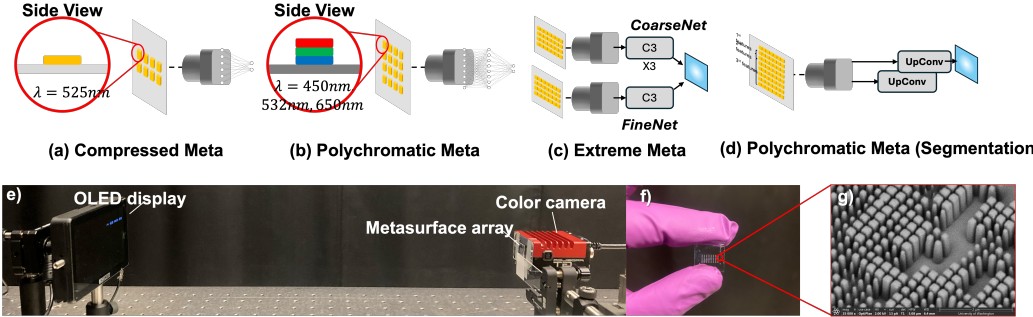

Figure 3: Optical systems. (a) Compressed Meta ONN on MNIST [37]; (b) Polychromatic Meta ONN on CIFAR-10 [29]; (c) ExtremeMETA for segmentation with dual optical frontends [5]; (d) Customized Polychromatic Meta ONN for segmentation (Ours); (e) Optical measurement setup; (f) Fabricated PSF-engineered meta-optics; (g) Scanning electron microscopy image of the meta-optics.

Table 2: Teacher and student network details.

| Student Architecture | Teacher Model | Teacher Accuracy | Teacher Size |
|---|---|---|---|
| Compressed Meta (0.05M) | LeNet | 99.1% | ∼0.18M |
| Polychromatic Meta (1.62M) | AlexNet | 85.4% | 233.29M |
| Polychromatic Meta (1.06M) | U-Net | 95.4% | 196.97M |

noise. Indeed, if $m \gg \|\delta\|$, the impact of fabrication noise on the prediction can be considered negligible. If $\delta$ is interpreted as a gradient descent step, Eq. 11 is consistent with NTK theory: as $m \to \infty$, the NTK ($\Theta$) remains constant during training, implying $\Delta\Theta \to 0$ [57].

To correct these errors, we re-apply minimization in Eq. 9, but restrict optimization to the unfrozen backend parameters (shown in Figure 2.3). The teacher network takes the raw input images and computes its NTK matrix over a batch of samples, as defined in Eq. 7. The student network receives feature maps from the fixed optical frontend, which processes the same batch of input images and produces an NTK matrix of size $n_{\text{batch}} \times n_{\text{batch}}$.

## 4 Results

### 4.1 Implementation Details, Pre-trained Teachers, Datasets and Evaluation Metrics

**Implementation Details:** Figure 3 demonstrates four different optical systems used in the experiments. For monochromatic image classification, we conducted experiments on the Compressed Meta ONN architecture [37] using the MNIST dataset [59]. This system consists of a single optical frontend with 8 kernels ($7 \times 7$) and a compact digital backend composed of two fully connected layers. For polychromatic image classification, we evaluated the Polychromatic Meta ONN [29] on the CIFAR-10 dataset [60]. This model consists of a single optical frontend with 16 kernels ($7 \times 7$) and a digital backend consisting of three fully connected layers.

For image segmentation, we performed experiments on both the Extreme Meta ONN [5] and a modified version of the Polychromatic Meta ONN [29], using Kaggle's Carvana Image Masking dataset. ExtremeMETA consists of two parallel polychromatic optical frontends, followed by a dual-path digital backend composed of CoarseNet for global feature extraction and FineNet for detail enhancement, with their outputs fused to produce the final segmentation. Our customized polychromatic segmentation system extends the Polychromatic Meta ONN by incorporating a single optical frontend with 56 kernels (8, 16, and 32 kernels to capture hierarchical depth representations, each of size $3 \times 3$) and a backend composed of upconvolutional layers to support dense prediction tasks. For a fair comparison between the two systems, we matched the number of optical kernels. Optical implementation details are in the Supplementary.

In experiments, we manipulate polychromatic−red, green, and blue−point spread functions (PSFs) of the meta-optics using a gradient descent algorithm. Physical shapes and dimensions of the PSF-

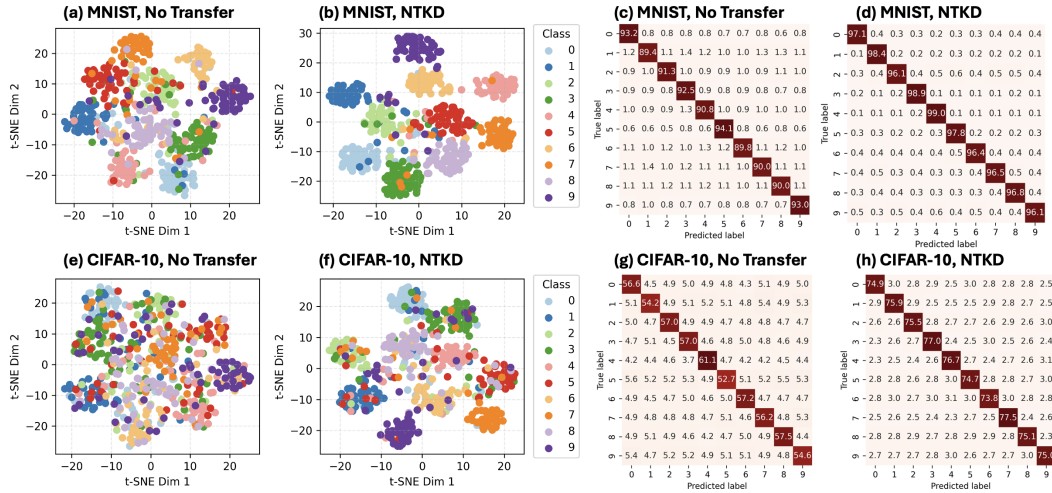

Figure 4: Simulation: t-SNE and confusion matrices in MNIST (a-d) and CIFAR-10 (e-h).

engineered meta-optics are shown in Figure 3. Since the meta-optics are designed with PSFs that function as convolutional kernels, optical convolution occurs naturally during image capture. Our experimental setup is straightforward: we replace a conventional imaging lens with PSF-engineered meta-optics. The meta-optics are carefully positioned and aligned with the color camera, enabling the capture of PSFs and convolved images using a laser pointer and an OLED display, respectively. A schematic representation and a photograph of the setup are provided in Figure 3.

Computing the NTK explicitly via Jacobian–Jacobian products is memory-intensive and infeasible at scale. To address this, we adopt the approximation strategy of NTK-SAP [61–63], which estimates the NTK trace rather than constructing the full matrix. We use batch size of 128 for MNIST and CIFAR-10/100, batch size of 64 for ImageNet-100, and batch size of 8 for segmentation tasks.

**Pre-trained teachers and Datasets:** The MNIST and CIFAR-10 datasets each consist of $50,000$ training images and $10,000$ testing images. The Carvana dataset, originally introduced in Kaggle's Carvana Image Masking Challenge, contains 5,088 high-resolution $1920 \times 1280$ car images. We adopt LeNet (99.1% accuracy on MNIST), AlexNet (84.5% on CIFAR-10), and a full U-Net (95.4% mIoU on Segmentation) as teacher models for their respective tasks. Table 2 summarizes the teacher and student networks.

**Evaluation Metrics:** For classification tasks, we ran each experiment five times with different random seeds and reported the mean and standard deviation (std) of the classification accuracy. For segmentation tasks, we similarly conducted five independent runs and reported the mean Intersection over Union (mIoU) along with the standard deviation.

## 4.2  Main Results

**Simulation Results:** Table 3 summarizes the accuracy of different ONN training strategies in classification and segmentation tasks. For monochromatic classification, NTKD achieved 97.3% accuracy and outperformed both KD-based transfer (95.9%) and the non-transfer baseline (91.4%). Similar trends were observed in the more challenging polychromatic classification setting, where NTKD achieved 75.6%, surpassing KD (72.5%) and baseline (56.4%). For segmentation tasks, we evaluated models on both the Extreme Meta and our proposed Polychromatic Meta datasets. NTKD

Table 3: Performance comparison of ONN methods across different tasks and training strategies.

| Methods | Monochromatic Classification (%) Compressed Meta (2025) [37] | Polychromatic Classification (%) Polychromatic Meta (2025) [29] | Polychromatic Segmentation (mIoU) Extreme Meta (2025) [5] | Polychromatic Meta (Ours) |
|---|---|---|---|---|
| Simulation, No Transfer | $91.4 \pm 0.8\%$ | $56.4 \pm 1.9\%$ | $68.3 \pm 0.5\%$ | $74.3 \pm 0.4\%$ |
| Simulation, KD | $95.9 \pm 0.6\%$ | $72.5 \pm 2.1\%$ | $75.3 \pm 0.2\%$ | $86.7 \pm 0.4\%$ |
| Simulation, NTKD | $97.3 \pm 0.6\%$ | $75.6 \pm 0.9\%$ | $80.1 \pm 0.2\%$ | $91.5 \pm 0.4\%$ |

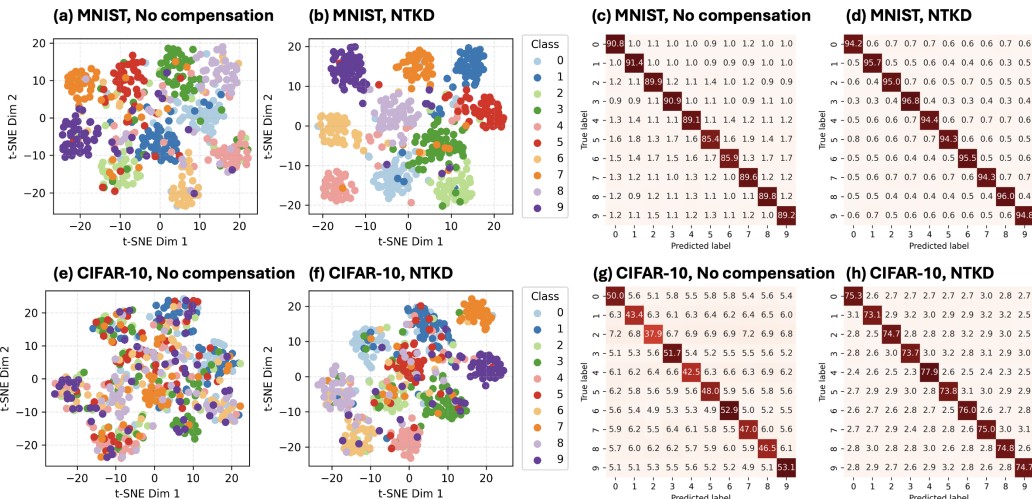

Figure 5: Fabrication: t-SNE and confusion matrices on MNIST (a–d) and CIFAR-10 (e–h).

consistently achieved higher mIoU scores across both optical systems, surpassing KD-based transfer and end-to-end training without transfer (75.3% and 86.7%, respectively).

These results indicate that training ONNs end-to-end without transfer often leads to suboptimal performance. Incorporating knowledge transfer through KD or NTKD improves learning efficacy and overall segmentation quality. In particular, the NTKD approach outperformed KD in different tasks, demonstrating its ability to guide representation learning in optical neural networks.

Figure 4 demonstrates knowledge transfer strategies on MNIST and CIFAR-10 representations and classification performance. Compared to the no-transfer baseline, these strategies improve class separability in t-distributed Stochastic Neighbor Embedding (t-SNE) visualizations and reduce noise in confusion matrices. NTKD transfer shows improved clustering and accuracy in experiments.

**Fabrication and Compensation Results:** Table 4 summarizes the impact of error compensation strategies on ONN performance in classification and segmentation tasks. Due to unavoidable fabrication and experimental errors, optical frontends suffer significant accuracy drops, especially in the polychromatic setting. Without compensation, the monochromatic system exhibits an 8.1% drop, and the more fabrication-sensitive polychromatic system shows a 28.3% drop on CIFAR-10 and a 41.8% reduction in mIoU on the Carvana segmentation task. This performance gap highlights the increased challenge of fabricating RGB-sensitive kernels in polychromatic ONNs compared to monochromatic ONNs. Our results demonstrate that knowledge transfer methods are able to assist with denoising that gap. NTKD compensation yields higher accuracy in both cases (95.1% for monochromatic, 74.9% for polychromatic and 81.2% for image segmentation task), outperforming end-to-end deep learning compensation, and validating its effectiveness in robust corrections for fabrication.

Figure 5 compares the t-SNE visualizations and confusion matrices of MNIST and CIFAR-10 representations under different compensation strategies. Without compensation (Figures 5 a, c, e, g), both optical systems exhibit class overlap in the feature space and reduced classification accuracy, primarily due to fabrication errors and optical misalignments. The NTKD correction (Figures 5 b, d, f, h) compensates for these errors and improves the clustering structure and classification accuracy, demonstrating robustness and generalization across datasets and optical systems.

Table 4: Evaluation of ONN error compensation methods on fabricated optical systems across both classification and segmentation tasks.

| Method | Monochromatic Classification (%) Compressed Meta (2025) [37] | Polychromatic Classification (%) Polychromatic Meta (2025) [29] | Polychromatic Segmentation (mIoU) Polychromatic Meta (Ours) |
|---|---|---|---|
| No compensation (baseline) | 89.2% | 47.3% | 49.7% |
| Error compensation (End-to-End) | 93.2 ± 0.1% | 70.4 ± 2.1% | 62.7 ± 0.9% |
| Error compensation (NTKD) | 95.1 ± 0.1% | 74.9 ± 1.3% | 81.2 ± 0.6% |

Table 5: Random PSF kernel design across different tasks.

| | Monochromatic Classification (%) | Polychromatic Classification (%) | Polychromatic Segmentation (mIoU) |
|---|---|---|---|
| 8 kernels | 96.12% | 36.73% | 49.3% |
| 500 kernels | 96.81% | 49.32% | 64.1% |
| 1000 kernels | 97.24% | 56.23% | 69.9% |
| $\infty$ kernels | 97.72% | 67.33% | 72.1% |

## 4.3 Discussion and Ablation Study

**Backend Complexity of ONNs:** The complexity of the backend plays a critical role in the performance of optical systems, particularly in hybrid optical-digital architectures. When a strong digital backend is employed, it can effectively denoise and recover accurate outputs, even when the optical frontend introduces significant noise. However, a strong backend not only diminishes the contribution of the optical frontend but also significantly increases power consumption—undermining the core motivation for adopting optical computing in resource-constrained environments. In such scenarios, digital networks (e.g., ViT or U-Net) are often a more practical and effective choice than hybrid ONNs with disproportionately strong backends. For practical deployment, we need to carefully balance the computational load between optics and computational backend and find a tradeoff between acceptable energy consumption/ latency and acceptable accuracy, which will be an application dependent trade-off.

**Random vs. Designed Parameters:** Another possible direction for designing and training ONNs is to use randomly initialized optical parameters while training only the digital backends. This approach aims to avoid the need for extensive simulation and hardware-in-the-loop optimization of the optical frontends. We conducted experiments using a single optical convolutional layer and a lightweight backend—consisting of a single fully connected layer for classification, or a single upsampling layer for segmentation. As shown in Table 5, increasing the number of random kernels consistently improves performance across tasks. For example, in polychromatic classification, accuracy improves from 36.73% (8 kernels) to 56.23% (1000 kernels), and further to 67.33% in the NTK regime, which approximates an infinite number of random kernels. Similarly, in polychromatic segmentation, mIoU rises from 49.3% to 69.9% and reaches 72.1% under NTK estimation. These results demonstrate that while increasing the number of random PSFs improves performance, they still underperform our approach (designed kernels with knowledge transfer).

**Scalability of ONNs:** Scaling current ONNs remains challenging, as most designs rely on shallow structures with limited linear computational capacity. Implementing nonlinear operations in ONNs is especially difficult due to physical constraints, such as the limited pixel size. These hardware limitations make it hard for ONNs to support deep and expressive architectures like those used in digital networks. We observed that using different kernels to simulate multiple layers of a digital network leads to better performance, compared to simply compressing a deep CNN into a single-layer ONN.

Table 6: Impact of Teacher Complexity on NTK Distillation Performance for Classification and Segmentation Tasks.

| Dataset | Task | Teacher | Student | Teacher Accuracy | Student Accuracy (with / without NTKD) |
|---|---|---|---|---|---|
| ImageNet-100 | Classification | ResNet-18 | Polychromatic Meta | 78.43% | 46.32% / 33.45% |
| ImageNet-100 | Classification | ResNet-50 | Polychromatic Meta | 88.32% | 47.86% / 33.45% |
| COCO-Stuff 10k | Segmentation | U-Net | Polychromatic Meta | 61.89% | 41.43% / 35.03% |
| COCO-Stuff 10k | Segmentation | ResU-Net | Polychromatic Meta | 69.23% | 42.73% / 35.03% |

Table 6 examines the impact of stronger teachers on the student ONN under optical physical limitations, with additional experiments conducted using ResNet variants and more complex datasets such as ImageNet-100 and COCO-Stuff 10k. For the classification task, we employed both ResNet-18 and ResNet-50 to train a Polychromatic Meta student network. While both teacher models improved performance through NTK distillation, the gain from ResNet-18 to ResNet-50 was marginal with only an increase of 1.54%. Similarly, for segmentation, we used U-Net and ResU-Net as teacher networks to train a Polychromatic Meta-optical student on COCO-Stuff 10k for binary foreground-background segmentation. In this setting, the foreground includes all semantic object classes, and the background consists of non-object regions. Again, while both stronger teacher models provided improvements, the performance gain from a more complex teacher was limited.

These results indicate that bottleneck in performance is primarily due to the physical modeling limitations of current ONN hardware, rather than the complexity of the teacher model. In the case of ONN technology being improved in the future, as one would expect, in such a case, a stronger teacher network can provide additional gains in performance through the NTKD approach that will transfer knowledge from stronger teachers to enhanced students. In summary, scalable and expressive ONNs will ultimately rely on physical advances enabling deep and nonlinear optical computations.

**Fabrication Analysis:** Several factors may help explain the discrepancy between the designed and measured kernels. First, the local periodic approximation, which simplified the metasurface optics design by assuming the scatterers were arranged periodically, neglected the coupling between adjacent dissimilar scatterers and could introduce phase errors. Second, unavoidable fabrication errors further contributed to the observed discrepancy. Third, the pre-designed kernel needed to be properly matched to the sensor's pixel array; any misalignment between the metasurface optics and the camera could also lead to deformation of the measured kernels. Regarding the polychromatic versus single-wavelength kernels, metasurfaces inherently suffered from strong chromatic aberrations, a universal characteristic of diffractive optics. While we co-optimized the kernel across multiple wavelengths during the design process to ensure consistent behavior, the performance remained limited by the intrinsic material properties.

**MACs and Power Consumption:** We estimated the multiply–accumulate operations (MACs) and power consumption of hybrid ONNs using our polychromatic ONN as an example (details in the Supplementary). The total number of MAC operations in the simulated digital network is approximately 239 MMACs (while the full U-Net requires 65.9 GMACs and Efficient U-Net reaches 1.37 GMACs), which is reduced to 65 MMACs after incorporating optical frontends [64]. The total energy consumption includes both image capture and digital computation. The full U-Net (pre-trained teacher) consumes 2.03 J for computation and 2.36 mJ for image capture, totaling **2.04 J** per image. The compact digital network requires 7.37 mJ for computation and 2.36 mJ for image capture, totaling **9.73 mJ** per image. In contrast, our hybrid ONN consumes 3.82 mJ for image capture and 2.01 mJ for backend processing, totaling **5.83 mJ**—representing over a 40% reduction in system-level energy consumption compared to the simulated digital network, and over $300\times$ energy compression compared to the pre-trained teacher U-Net.

## 5   Conclusion

We propose a comprehensive NTKD pipeline that addresses multiple tasks and multiple optical systems in ONN design, training, and compensation. By incorporating knowledge transfer, particularly NTK-based knowledge distillation, our framework consistently improves the accuracy of different optical systems across both classification and segmentation tasks. Based on extensive experiments, we observe that the current ONN performance is primarily limited by the shallow and linear nature of existing optical architectures. Future advances in deeper, nonlinear ONNs may help narrow the performance gap between optical and electronic neural networks, improving scalability and accuracy.

### Acknowledgement

The research is supported by the National Science Foundation (EFRI-BRAID-2223495) and partial support from HDR Institute: Accelerated AI Algorithms for Data-Driven Discovery (A3D3) National Science Foundation grant PHY-2117997 (JX,ES). Part of this work was conducted at the Washington Nanofabrication Facility/ Molecular Analysis Facility, a National Nanotechnology Coordinated Infrastructure (NNCI) site at the University of Washington with partial support from the National Science Foundation via awards NNCI-1542101 and NNCI-2025489. The authors also acknowledge the partial support by the Departments of Electrical Computer Engineering, Applied Mathematics and Physics at the University of Washington. The authors are also thankful to the eScience Center at the University of Washington.

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
