# OpenReview forum: "Neural Tangent Knowledge Distillation for Optical Convolutional Networks"
_NeurIPS.cc/2025/Conference — NeurIPS 2025 poster_

### Official Review · Reviewer_KgM7 · 2025-07-02

**Clarity:** 2
**Significance:** 3
**Originality:** 3
**Rating:** 5
**Confidence:** 4

**Summary:**

This work introduces a unified framework for ONNs that integrates the Neural Tangent Knowledge Distillation strategy to address key limitations in current ONN research: 1) the accuracy gap between ONNs and large-scale digital neural networks during training; 2) the performance degradation caused by discrepancies between simulation and physical fabrication, stemming from optical misalignments, material imperfections, and sensor noise. The extensive experiments show that the proposed pipeline works for both image classification and segmentation tasks.

**Questions:**

Please check my questions in the weaknesses part.

**Ethical Concerns:**

["NO or VERY MINOR ethics concerns only"]

**Final Justification:**

The authors have adequately addressed my concerns regarding the dataset and teacher network selection. Their clarifications were clear and convincing. I believe the proposed approach has strong practical value for ONN applications. Based on the rebuttal and improvements, I have decided to raise my rating.

**Limitations:**

Yes

**Quality:**

3

**Strengths And Weaknesses:**

Strengths:

S1. The motivation is clear and the introduction of NTKD seems clever.

S2. The method demonstrates good performance on segmentation tasks, which are particularly challenging for optical neural networks due to their complexity and dense prediction requirements.

S3. The proposed method is validated on both simulated and fabricated systems, bridging simulation-reality gaps.

Weaknesses:

W1. The NTK-based estimation in this work appears to be demonstrated only for PSF-based implementations, and it remains unclear how it generalizes to other optical systems like 4f architectures. This raises some concern regarding the hardware-agnostic claim of the proposed pipeline.

W2. From line 110, the authors state that “The optical convolution is realized either through a 4f system or via point spread function (PSF)-based free-space propagation.” However, the paper does not present any experimental results involving a 4f system. This omission undermines the claim of the pipeline being hardware-agnostic.

W3. The experiments for classification tasks are limited to MNIST and CIFAR-10, both of which are relatively simple datasets. To more convincingly demonstrate the generalizability and robustness of the proposed approach, it would be beneficial to evaluate on more complex datasets, such as CIFAR-100 or ImageNet subsets.

W4. The criteria for selecting teacher networks are not clearly explained. The authors mention in line 145 that they “assume access to a pretrained teacher network, such as LeNet for MNIST, AlexNet for CIFAR-10, or U-Net for image segmentation tasks.” However, it is unclear why more powerful or modern teacher networks were not considered, which might further improve distillation effectiveness and the student ONN’s performance.

---

> ### Author Rebuttal · Authors · 2025-07-30
>
> We sincerely thank the reviewer for their feedback and recognizing the motivation, novelty of our NTKD approach, and its effectiveness in both classification and segmentation tasks, as well as its potential to bridge the simulation-to-fabrication gap in ONNs.
>
> ## 1. Clarification on Optical System Generality (W1–2)
>
> We appreciate the reviewer’s concern regarding the inclusion of 4f optical systems, which may seem to challenge the “hardware-agnostic” claim of our pipeline. We would like to clarify that this concern likely stems from a conflation of theoretical feasibility and practical implementation scope. **To clarify this, we address the issue in two parts: (1) theoretical equivalence of optical convolutions; and (2) practical motivation for implementing PSF-based ONNs.**
>
> ### 1) **Theoretical equivalence of optical convolutions**
>
> Both the 4f optical system and PSF-engineered metasurface follow the convolutional relationship as the multiplications in the frequency space are identical to the convolutional operations in the real space [1]. Once convolutional kernels are trained digitally, the convolution operation can be physically implemented by either a 4f optical system or a PSF-based metasurface. Therefore, we state that “the optical convolution is realized either through a 4f system or via PSF-based free-space propagation.”, which is supported by prior studies [1–4].
>
> Specifically, the 4f system can realize convolution operation by performing a 2D Fourier transform of the input image using the first lens by Fraunhofer diffraction, applying a spatial frequency filter in the Fourier plane, and then using the second lens to perform an inverse Fourier transform [1]. In our context, once a convolutional kernel is trained by NTKD, it can be converted into an optical filter design by mapping its frequency-domain representation onto the spatial light modulator or mask placed at the Fourier plane.
>
> ### **2) Practical Implementation Choice**
> While we acknowledge that our experimental implementation focuses on the PSF-based system, this decision is guided purely by practicality rather than technical limitation. Previous works have identified several challenges in scaling 4f systems for real-world deployment:
> - **Bulk and alignment sensitivity**: A 4f system typically requires two lenses and a spatial filter, increasing the physical size and introducing greater sensitivity to misalignments [5].
>
> - **Parallelization bottlenecks**: In 4f systems, the filtering optics must be confined to a compact region at the Fourier (focal) plane, which restricts the system’s ability to perform parallel processing. This architectural constraint makes it challenging to efficiently handle multi-channel inputs (e.g., RGB). In contrast, metasurfaces can modulate the entire optical wavefront in a single chip, providing a more compact, scalable, and multiple-channel solution [5-6].
>
> In summary, our physical prototype adopts the PSF-based metasurface design to reflect a more realistic and deployable architecture, while maintaining theoretical compatibility with 4f architectures. **To address the reviewer’s concern, we will also adjust our claim in the original paper:**
>
> _"Prior works have demonstrated that optical convolution can be physically realized using either a 4f system or a PSF-based free-space propagation system [1-4]. "_
>
> _"In this work, we practically implement a PSF-based metasurface design due to its advantages in compactness, alignment robustness, and ease of fabrication [5-6]."_
>
> ## 2. Clarification on Dataset and Teacher Network Selection (W3–4)
> We appreciate the reviewer’s feedback and agree that our current experiments are based on relatively simple datasets (e.g., MNIST, CIFAR-10) and teacher networks (e.g., AlexNet).
>
> This choice reflects the current state of the field, particularly for **SOTA ONN architectures** such as Polychromatic Meta (Nature Communications 2025) [1], where the primary bottleneck lies in the complexity of physical implementation. Most fabricated ONNs are limited to shallow and linear architectures due to challenges in optical alignment, lack of physical nonlinear activation, and maintaining signal fidelity. Given these optical constraints, our goal is not to match the performance of large-scale digital networks, but to improve ONN accuracy under realistic hardware limitations. Our pipeline is designed to enhance ONN performance in such settings and help narrow the gap between ONNs and digital networks. In summary, while the datasets in our experiments are considered basic in vision tasks, they remain state of the art benchmark datasets in the ONN field.
>
> **To address the reviewer’s concern and to examine the impact of stronger teachers on the student ONN according to optical physical limitations that we describe above and in the paper, we also conducted additional experiments using ResNet variants and more complex datasets, including ImageNet-100 and CIFAR-100.** For the classification task (as we stated in the main paper, line 277), we employed both ResNet-18 and ResNet-50 to train a Polychromatic Meta student network. While both teacher models improved performance through NTK distillation, the gain from ResNet-18 to ResNet-50 was marginal, with only a 1.54% increase.
>
> These results indicate that the performance bottleneck is primarily due to the physical modeling limitations of current ONN hardware, rather than the complexity of the teacher model. We do expect the ONN technology to improve in the future, where a stronger teacher network can provide added benefit. In such cases the NTKD approach should be applicable for knowledge transfer to enhanced students. We will add the results below to the revised version of our paper and include the discussion above in the discussion section.
>
> **Table 1: Impact of Different Teacher Models and NTKD on Student Performance**
>
> | Dataset        | Teacher     | Student             | Student Accuracy (with / without NTKD) |
> |----------------|-------------|---------------------|----------------------------------------|
> | ImageNet-100   | ResNet-18   | Polychromatic Meta  | 46.32% / 33.45%                        |
> | ImageNet-100   | ResNet-50   | Polychromatic Meta  | 47.86% / 33.45%                        |
> | CIFAR-100      | ResNet-18   | Polychromatic Meta  | 49.32% / 43.37%                        |
> | CIFAR-100      | ResNet-50   | Polychromatic Meta  | 49.54% / 43.37%                        |
> |||||
>
> [1] Chang, Julie, et al. "Hybrid optical-electronic convolutional neural networks with optimized diffractive optics for image classification." Scientific reports 8.1 (2018): 12324.
>
> [2] Cutrona, Laj, et al. "Optical data processing and filtering systems." IRE Transactions on Information Theory 6.3 (2003): 386-400.
> [3] Burgos, Carlos Mauricio Villegas, et al. "Design framework for metasurface optics-based convolutional neural networks." Applied Optics 60.15 (2021): 4356-4365.
>
> [4] Xiang, Jinlin, et al. "Knowledge distillation circumvents nonlinearity for optical convolutional neural networks." Applied Optics 61.9 (2022): 2173-2183.
>
> [5] Wirth-Singh, Anna, et al. "Compressed meta-optical encoder for image classification." Advanced Photonics Nexus 4.2 (2025): 026009-026009.
>
> [6] Choi, Minho, et al. "Transferable polychromatic optical encoder for neural networks." Nature Communications 16.1 (2025): 5623.

---

> > ### Comment · Reviewer_KgM7 · 2025-08-05
> >
> > Thank you for your thoughtful feedback. Your response addresses most of my concerns. I will update my rating accordingly.

---

> > > ### Author Response · Authors · 2025-08-05
> > >
> > > Thank you for taking the time to consider our response and provide your update. We truly appreciate your review.

---

### Official Review · Reviewer_zR5Y · 2025-07-02

**Clarity:** 3
**Significance:** 3
**Originality:** 3
**Rating:** 5
**Confidence:** 4

**Summary:**

This paper introduces a neural tangent kernel (NTK)-based knowledge distillation (NTKD) method to enhance the performance of hybrid opto-electronic neural networks (ONNs). Through extensive simulations and real-world experiments in image classification and segmentation, the authors demonstrate that NTKD significantly improves model accuracy while compensating for fabrication errors via digital-backend fine-tuning.

**Questions:**

Please refer to [Cons]

**Ethical Concerns:**

["NO or VERY MINOR ethics concerns only"]

**Final Justification:**

I am satisfied with the authors' response. I thus raise my score and vote for acceptance of this work.

**Limitations:**

yes

**Quality:**

3

**Strengths And Weaknesses:**

## [Pros]
The major novelty of this paper lies at the proposed NTKD loss defined in (8), for which the authors demonstrate via comprehensive experiments the advantages over existing baselines. The effectiveness is evaluated under various optical neural network settings (monochromatic, polychromatic, different architectures, datasets) in both simulation and real-world prototype experiments, which is in general convincing.

## [Cons]

### Major comments:
As a seemingly generic knowledge distillation method, the authors should discuss why this is tailored for optical neural networks (ONNs). NTK, as a (somewhat impractical) theoretical tool is seldom used in deep learning practice as the large computation efforts are entailed to evaluate the NTK matrices. The use of NTKD loss function should be highly expensive, thus hindering its applicability in deep neural networks. It's therefore unclear how did authors apply this loss function to relative heavy deep neural network (i.e., U-Net as the teacher network mentioned for image segmentation)
The dimension specification of Jacobians matrices in L149 seems to be wrong. The number of classes is ignored there, IMO, the Jacobians matrix dimension should be $n_{batch} \times p*c$ where c is the number of classes. Such a linear dependency in number of classes also imposes larger computation burden for handling more complex image classification tasks (such as ImageNet classification)

### Minor comments:
In Table 2, the results of the teacher networks should be added as references. The number of digital parameters of the teacher and student network should also be provided for comprehensive evaluation of the proposed method. The paragraph "Parameter-Space Compressibility" in Section 4.3 somewhat deviates from central theme of this paper, thus would be better defered to supplementary material. Similarly, more discussions of the ONNs in supplementary could be moved in main paper.
In abstract, the authors state "To assist optical system design before training, we estimate achievable model accuracy based on user-specified constraints such as physical size and the dataset", however, there is no insightful discussion in the paper how the NTK can be used to guide ONN design before training.

### Justification
Through the proposed NTKD method, this paper advances a recent line of research to bridge the gap between ONN and electronic networks. The proposed method looks sound and the experimental results are promising. However, the computation cost and the scalability are not well exposed, which are likely the key limitations of the proposed approach. As a result, I hold a borderline opinion at this moment, and would adjust my position in the rebuttal stage.

---

> ### Author Rebuttal · Authors · 2025-07-30
>
> We appreciate the reviewer’s thoughtful feedback and the recognition of our NTK-based loss and the comprehensive experimental validation across diverse ONN architectures and tasks. While the NTK loss is an important component, our main contribution lies in proposing a holistic pipeline for ONNs, encompassing training, fabrication, and post-fabrication fine-tuning. Motivated by the challenges of bridging simulated and fabricated ONNs, we demonstrate that NTK-based distillation provides a general and effective solution. Our pipeline is validated across multiple ONN designs, including empirical optical implementations, to systematically improve ONN performance under hardware constraints and narrow the gap to digital networks.
>
> ## 1. Clarification on NTKD Applicability and NTK Computation Feasibility (Major Concern)
> > “As a seemingly generic knowledge distillation method, the authors should discuss why this is tailored for optical neural networks (ONNs).”
> ### **Clarification on NTKD’s Suitability for ONNs:**
> We appreciate the reviewer’s concern and would like to clarify why our NTK-based distillation is particularly suited for optical neural networks (ONNs). Most SOTA ONNs use optical frontends that perform fixed and linear operations, such as diffraction or Fourier transforms. NTK provides a linear approximation of how the network behaves under small parameter changes, which aligns well with the linear nature of our optical systems. This makes NTK-based matching a natural fit for transferring knowledge from digital teacher models to ONNs.
>
> ### **Clarification on NTK calculation:**
> Explicit computation of the NTK via Jacobian-Jacobian products is indeed memory-intensive and could become impractical for large-scale settings, especially when scaling to deeper networks or larger datasets such as ImageNet or COCO. We also thank the reviewer for pointing out the omission of the number of classes in the Jacobian dimension specification (L149). We will revise the text.
>
> To make NTK computationally feasible across different network architectures and scales, we follow the approximation strategy proposed in NTK-SAP (ICLR 2023) [2]. This method has been shown to be effective in computing NTK for ResNet-18 and ResNet-50, further supporting its scalability to deep architectures. Specifically, this method estimates the trace of the NTK rather than constructing the full NTK matrix, defined as
>
> $\|\| \Theta \|\|_{tr} = \|\| \nabla _\theta f(X; \theta) \|\|^2_F$
>
> where $\Theta$ denotes the Neural Tangent Kernel (NTK), $f(X; \theta)$ represents the neural network output for an input batch and parameters $ \theta $, and $ \|\| \cdot \|\|^2_F $ denotes the squared Frobenius norm.
>
> $
> \|\|\nabla_\theta f(X; \theta)\|\|^2_F \approx \frac{1}{\epsilon} \cdot \mathbb{E}_{\Delta\theta \sim \mathcal{N}(0, \epsilon I)} \left[\|\| f(X; \theta) - f(X; \theta + \Delta\theta)\|\|^2_2\right]
> $
>
> where we set $\epsilon$=0.01, following the choice in the original NTK-SAP paper. The computational load is comparable to a single forward pass in evaluation mode. While it is an approximation, this strategy has been validated in prior work (NTK-SAP, ICLR 2023) to capture essential NTK trends and ensure stable training, making it a practical choice for scaling NTK-based methods to deeper networks and realistic ONN applications [2-3].
>
> **To address the reviewer’s concern, we have added the following clarification to the main paper**:
>
> _"Explicitly computing the NTK via Jacobian–Jacobian products is memory-intensive and impractical for large-scale settings. Therefore, we follow the approximation strategy proposed in NTK-SAP (ICLR 2023), which estimates the trace of the NTK rather than constructing the full NTK matrix."_
>
> ### **NTK Distillation Load Analysis in Image Segmentation (U-Net):**
> In our segmentation task, the input is a 3-channel image of size H×W (shape: [B, 3, H, W]), and the model outputs a predicted binary mask with shape [B, 1, H, W]. Following the definition in NTK-SAP, the Jacobian matrix has a shape [B × H × W, number_of_parameters]. We adopt this reshaping convention (as described in Section 4, Equations 4–6 of NTK-SAP) to calculate the NTK trace for image segmentation.
>
> ## 2.Clarification on Minor Comments
> We thank the reviewer for the helpful suggestions and agree with all the points raised.
> > "In Table 2, the results of the teacher networks should be added as references. The number of digital parameters of the teacher and student network should also be provided for comprehensive evaluation of the proposed method. "
>
> - The teacher model results are already included in Lines 203–207; we will also add this information to Table 2 for clarity.
> **Table: Comparison of NTKD Performance Across Different Student Architectures and Teacher Models**
>
> | Student Architecture            | Teacher Model        | Teacher Accuracy | Teacher Size  |
> |--------------------------------|----------------------|------------------|---------------|
> | Compressed Meta (0.05M)        | LeNet                | 99.1%            | ~0.18M        |
> | Polychromatic Meta (1.62M)     | AlexNet              | 85.4%            | 233.29M       |
> | Polychromatic Meta (1.06M)     | U-Net                | 95.4%            | 196.97M       |
> |||||
>
> > "The paragraph "Parameter-Space Compressibility" in Section 4.3 somewhat deviates from central theme of this paper, thus would be better defered to supplementary material. Similarly, more discussions of the ONNs in supplementary could be moved in main paper. "
>
> - The paragraph on “Parameter-Space Compressibility” will be moved to the supplementary material. We will also add discussion from the supplementary material as suggested.
>
> >"In abstract, the authors state "To assist optical system design before training, we estimate achievable model accuracy based on user-specified constraints such as physical size and the dataset", however, there is no insightful discussion in the paper how the NTK can be used to guide ONN design before training."
>
> - We thank the reviewer for pointing out the need for further clarification of the abstract. While the NTK provides a theoretical estimate of model capacity and can be used to predict performance trends before training (e.g., Eq.4 in main paper: $f(x^{\text{test}};\theta) = \Theta_{\text{test,train}} \left( \Theta_{\text{train,train}} + \lambda I \right)^{-1} y^{\text{train}}$), we agree that its utility for guiding ONN design is less practical in real-world scenarios, particularly when applied to large-scale datasets.
>
> **We will revise the abstract to more accurately reflect our contribution:**
>
> _"To assist optical system design before training, we design the metasurface layout based on fabrication constraints."_
>
> [1] Wang, Y., Li, D., & Sun, R. (2023). NTK-SAP: Improving neural network pruning by aligning training dynamics. In Proceedings of the Eleventh International Conference on Learning Representations (ICLR).
>
> [2] Vogt, R., Zheng, Y., & Shlizerman, E. (2024). Lyapunov-guided representation of recurrent neural network performance. Neural Computing and Applications, 36(34), 21211-21226.
>
> [3] Zheng, C., & Shlizerman, E. (2025). Hyperpruning: Efficient Search through Pruned Variants of Recurrent Neural Networks Leveraging Lyapunov Spectrum. arXiv preprint arXiv:2506.07975.

---

### Official Review · Reviewer_MQJm · 2025-07-08

**Clarity:** 3
**Significance:** 3
**Originality:** 3
**Rating:** 5
**Confidence:** 4

**Summary:**

The paper proposes a Neural Tangent Kernel based knowledge distillation process to address the accuracy gap between Optical Neural Networks (ONNs) and large scale networks, and also the discrepancies between simulated and fabricated ONN performance. The paper considers ONN architectures that includes an optical frontend to perform early stage linear convolution and an electronic backend to perform fully connected non-linear operations. Once the optical frontend design is done, the ONN is trained in simulation with end-to-end loss and NTK based loss with pretrained and frozen teacher networks. After ONN fabrication, the digital backend is retrained with NTK loss to compensate for the differences in simulation and fabrication. Experiments are done using MNIST and CIFAR10 for classification, and Carvana for segmentation.

**Questions:**

1. How would the proposed NTK based approach scale for large models?
2. What’s the size of n_{batch} in the experiments and how is it determined?
3. How are the two loss terms balanced in Eq. 9 and how sensitive is this choice?

**Ethical Concerns:**

["NO or VERY MINOR ethics concerns only"]

**Final Justification:**

The paper proposes an effective approach for knowledge distillation for ONN and conducted experiments for different ONNs, network  architectures, and datasets. Scalability is a concern but the authors showed benefits even with an approximation startegy.

**Limitations:**

Yes

**Quality:**

3

**Strengths And Weaknesses:**

Strengths
-
1. The proposed NTK-based distillation approach is general enough for different ONN architectures and vision tasks.
2.The paper shows performance improvements on Compressed Meta, Polychromatic Meta, and Extreme Meta ONNs (in simulation) for both classification and segmentation.
3. Experimental evaluations are conducted in both simulated (pre-fabrication) and fabricated (post-fabrication) settings.

Weaknesses
-
1. The dataset and teacher networks used are not SOTA anymore and relatively simple. While these are used widely in ONN literature, using ResNet variants with ImageNet and COCO type datasets would’ve made the paper stronger.

2. NTK loss requires computing product of Jacobians which is computationally expensive and will not scale for large networks. Also, what’s the size of n_{batch} in the experiments and how is it determined?

3. The paper doesn’t include any discussion on alpha and beta in the Eq. 9. How are the two loss terms balanced?

---

> ### Author Rebuttal · Authors · 2025-07-30
>
> We appreciate the reviewer acknowledging the generalizability of our NTK-based distillation pipeline across diverse ONN architectures and vision tasks, as well as its contribution to advancing current ONNs’ performance in both simulated (pre-fabrication) and fabricated (post-fabrication) stages.
> Below, we provide clarifications in response to the reviewer’s questions.
>
> ## 1. Clarification on Dataset and Teacher Network Selection (W1)
> We appreciate the reviewer’s feedback and agree that our current experiments are based on relatively simple datasets (e.g., MNIST, CIFAR-10) and teacher networks (e.g., AlexNet).
>
> This choice reflects the current state of the field, particularly for **SOTA ONN architectures** such as Polychromatic Meta (Nature Communications 2025) [1], where the primary bottleneck lies in the complexity of physical implementation. Most fabricated ONNs are limited to shallow and linear architectures due to challenges in optical alignment, lack of physical nonlinear activation, and maintaining signal fidelity. Given these optical constraints, our goal is not to match the performance of large-scale digital networks, but to improve ONN accuracy under realistic hardware limitations. Our pipeline is designed to enhance ONN performance in such settings and help narrow the gap between ONNs and digital networks. In summary, while the datasets in our experiments are considered basic in vision tasks, they remain state of the art benchmark datasets in the ONN field.
>
> **To address the reviewer’s concern and to examine the impact of stronger teachers on the student ONN according to optical physical limitations that we describe above and in the paper, we also conducted additional experiments using ResNet variants and more complex datasets, including ImageNet-100 and COCO-Stuff 10k.** For the classification task (as we stated in the main paper, line 277), we employed both ResNet-18 and ResNet-50 to train a Polychromatic Meta student network. While both teacher models improved performance through NTK distillation, the gain from ResNet-18 to ResNet-50 was marginal, with only a 1.54% increase.
>
> Similarly, for the segmentation task, we used U-Net and ResU-Net as teacher networks to train a Polychromatic Meta-optical student on COCO-Stuff 10k for binary foreground-background segmentation. In this setting, the foreground includes all semantic object classes, and the background consists of non-object regions. Again, while both stronger teacher models provided improvements, the performance gain from a more complex teacher was limited.
>
> These results indicate that the performance bottleneck is primarily due to the physical modeling limitations of current ONN hardware, rather than the complexity of the teacher model. We do expect the ONN technology to improve in the future, where a stronger teacher network can provide added benefit. In such cases the NTKD approach should be applicable for knowledge transfer to enhanced students. We will add the results below to the revised version of our paper and include the discussion above in the discussion section.
>
> **Table 1: Impact of Teacher Complexity on NTK Distillation Performance for Classification and Segmentation Tasks**
>
> | Dataset         | Task          | Teacher     | Student             | Teacher Accuracy | Student Accuracy (with / without NTKD) |
> |----------------|---------------|-------------|----------------------|------------------|----------------------------------------|
> | ImageNet-100   | Classification| ResNet-18   | Polychromatic Meta  | 78.43%           | 46.32% / 33.45%                         |
> | ImageNet-100   | Classification| ResNet-50   | Polychromatic Meta  | 88.32%           | 47.86% / 33.45%                         |
> | COCO-Stuff 10k | Segmentation  | U-Net       | Polychromatic Meta  | 61.89%           | 41.43% / 35.03%                         |
> | COCO-Stuff 10k | Segmentation  | Res U-Net   | Polychromatic Meta  | 69.23%           | 42.73% / 35.03%                         |
> |||||||
>
> ## 2. Clarification on NTK calculation and batch size selection (W2, Q1-2)
>
> Explicit computation of the NTK via Jacobian-Jacobian products is indeed memory-intensive and could become impractical for large-scale settings, especially when scaling to deeper networks or larger datasets such as ImageNet or COCO.
>
> To make NTK computationally feasible across different network architectures and scales, we follow the approximation strategy proposed in NTK-SAP (ICLR 2023) [2]. This method has been shown to be effective in computing NTK for ResNet-18 and ResNet-50, further supporting its scalability to deep architectures. Specifically, this method estimates the trace of the NTK rather than constructing the full NTK matrix, defined as
>
> $\|\| \Theta \|\|_{tr} = \|\| \nabla _\theta f(X; \theta) \|\|^2_F$
>
> where $\Theta$ denotes the Neural Tangent Kernel (NTK), $f(X; \theta)$ represents the neural network output for an input batch and parameters $ \theta $, and $ \|\| \cdot \|\|^2_F $ denotes the squared Frobenius norm.
>
> Computing the Frobenius norm of the Jacobian matrix is both memory- and computation-intensive, especially for deep networks and large input batches. To alleviate this, NTK-SAP proposes a finite-difference approximation of the trace, which avoids explicit Jacobian construction. Following this approach, we approximate the NTK trace by computing the expectation
>
> $
> \|\|\nabla_\theta f(X; \theta)\|\|^2_F \approx \frac{1}{\epsilon} \cdot \mathbb{E}_{\Delta\theta \sim \mathcal{N}(0, \epsilon I)} \left[\|\| f(X; \theta) - f(X; \theta + \Delta\theta)\|\|^2_2\right]
> $
>
> where we set $\epsilon$=0.01, following the choice in the original NTK-SAP paper. The computational load is comparable to a single forward pass in evaluation mode. While it is an approximation, this strategy has been validated in prior work (NTK-SAP, ICLR 2023) to capture essential NTK trends and ensure stable training, making it a practical choice for scaling NTK-based methods to deeper networks and realistic ONN applications [3-4].
>
> **Batch Size**: We follow the batch size strategy from NTK-SAP on NVIDIA V100 GPUs, which uses batch sizes up to 256 for most datasets with ResNet-18 and ResNet-50. In our experiments, we observed that larger batch sizes lead to slightly higher loss, so we tested batch sizes of 64, 128, and 256 across different datasets. For U-Net segmentation with input size 384 by 216, we start with a batch size of 8 and also test smaller sizes such as 4 and 2. While the performance may slightly vary, the overall trends remain consistent, with accuracy fluctuations within ±2%.
>
> **To address the reviewer’s concern, we have added the following clarification to the main paper**:
>
> _"Explicitly computing the NTK via Jacobian–Jacobian products is memory-intensive and impractical for large-scale settings. Therefore, we follow the approximation strategy proposed in NTK-SAP (ICLR 2023), which estimates the trace of the NTK rather than constructing the full NTK matrix."_
>
> _"In terms of batch size, we use 128 for MNIST, CIFAR-10/100, 64 for ImageNet-100 experiments, and 8 for image segmentation tasks."_
>
> ## 3. Clarification on balancing loss terms (W3, Q3)
>
> We thank the reviewer for pointing this out. We experimentally observed that the two loss terms are on a similar scale, so we balance them using a scalar weight alpha, selected through hyperparameter search over {0.1, 0.3, 0.5, 0.7, 0.9} on a validation set. We will clarify this in the revised manuscript. For other datasets beyond those we tested, users can tune their own hyperparameters or design a custom decay schedule if needed.
>
> **To address the reviewer’s concern, we have added the following clarification to the main paper**:
>
> _"We experimentally observed that the two loss terms are on a similar scale, so we balance them using a scalar weight alpha, selected through hyperparameter search over {0.1, 0.3, 0.5, 0.7, 0.9} on a validation set."_
>
> [1] Choi, Minho, et al. "Transferable polychromatic optical encoder for neural networks." Nature Communications 16.1 (2025): 5623
>
>
> [2] Wang, Y., Li, D., & Sun, R. (2023). NTK-SAP: Improving neural network pruning by aligning training dynamics. In Proceedings of the Eleventh International Conference on Learning Representations (ICLR).
>
> [3] Vogt, R., Zheng, Y., & Shlizerman, E. (2024). Lyapunov-guided representation of recurrent neural network performance. Neural Computing and Applications, 36(34), 21211-21226.
>
> [4] Zheng, C., & Shlizerman, E. (2025). Hyperpruning: Efficient Search through Pruned Variants of Recurrent Neural Networks Leveraging Lyapunov Spectrum. arXiv preprint arXiv:2506.07975.

---

### Note · Authors · 2025-08-12

We sincerely thank the reviewers for their thoughtful feedback and constructive suggestions. We also appreciate the recognition of our work from the reviewers. In particular, Reviewer MQJm noted that “NTK-based distillation approach is general enough for different ONN” Reviewer zR5Y highlighted “the novelty of the proposed NTKD loss” and its evaluation “in both simulation and real-world prototype experiments, which is in general convincing,” and Reviewer KgM7 stated that “the motivation is clear,” the method “demonstrates good performance on segmentation tasks,” and is “validated on both simulated and fabricated systems, bridging simulation-reality gaps.”

Our major contributions are as follows:
- A general, novel NTKD pipeline supporting diverse tasks and optical structures, addressing challenges of shallow architectures and physical imperfections (R-MQJm, R-zR5Y, R-KgM7).
- Extensive validation on classification and segmentation tasks, showing improvements in both simulations and fabricated prototypes (R-MQJm, R-zR5Y, R-KgM7), with segmentation performance emphasized by R-KgM7.
- A metasurface layout design based on fabrication constraints to guide optical system design before training, with clarifications added per R-zR5Y’s request.

During the rebuttal, we addressed the following concerns:
- Dataset and teacher network choice: Fabricated ONNs are limited to shallow, linear architectures due to alignment challenges, lack of physical nonlinear activation, and the need to maintain signal fidelity. Our aim is to improve ONN accuracy under realistic hardware limits, not to match large-scale digital networks. Indeed, experiments that we added as a result of reviewers' feedback, i.e., experiments with ResNet-18/50 on ImageNet-100, COCO-Stuff 10k, and CIFAR-100, show only marginal gains from stronger teachers.
- Scalability and NTK computation cost: We clarified the use of NTK-SAP to approximate NTK computation efficiently, making it feasible for deeper networks and large datasets.
- Implementation choice: We clarified the theoretical equivalence between PSF and 4f systems and explained the practical challenges (e.g., bulk, alignment sensitivity, and parallelization bottlenecks) in 4f systems. Therefore, we adopted PSF in our prototype for practical reasons.

In summary, we have discussed these concerns with the reviewers, and they expressed satisfaction with our responses. We will incorporate all points mentioned in the rebuttal into the revised manuscript.

---

### Decision · Program_Chairs · 2025-09-17

**Decision:**

Accept (poster)

**Comment:**

Reviewers agree that this is a significant step forward in the field. The rebuttal and post-rebuttal discussion clarified initial questions. Congratulations, this submission is accepted.